# Software development of an internet-of-things based controlled-source ultra-audio frequency electromagnetic receiver

Xiyuan Zhang<sup>1</sup>, Qisheng Zhang<sup>1</sup>, Zucan Lin<sup>1</sup>, Huiying Li<sup>1</sup>, Xinchang Wang<sup>1</sup>, and Hui Zhang<sup>1</sup> School of Geophysics and Information Technology, China University of Geosciences, Beijing, China Correspondence: Oisheng Zhang (zqs@cugb.edu.cn)

Abstract. This study addresses the limitations of traditional CSAMT (Controlled-Source Audio-frequency Magnetotellurics) in insufficient shallow-to-medium layer exploration accuracy, complex human-machine interaction, and constrained data transmission in existing electromagnetic instruments. A novel IoT(Internet-of-Things)-integrated CSUMT-R (Controlled-Source Ultra-audio Frequency Electromagnetic Receiver) system is developed. The hardware employs a Zynq multi-core processor architecture, integrating a five-channel ultra-wide band (1Hz-1MHz) acquisition system. The software system features a dual-layer architecture combining embedded control and remote monitoring, incorporating dynamic buffer DMA (Direct Memory Access) drivers, distributed hybrid networking technology, IoT technology, and five-channel batch processing algorithms to support high-speed real-time data transmission (up to 320 Mbps) and remote visualization. Field tests in the Fengtai ore cluster area, Shaanxi Province, China, demonstrate stable functionality and high intelligence, meeting demands for complex field exploration.

#### 1 Introduction

20

6

Electromagnetic exploration technology serves as a cornerstone in geophysical surveys, playing pivotal roles in mineral exploration, geothermal resource assessment, hydrocarbon detection, and engineering geology (Guo et al., 2020; Constable & Srnka., 2007; Munoz, 2014; Xue et al., 2019). Among these methods, CSAMT, proposed by Goldstein & Strangway (1975), utilizes artificial sources to generate electromagnetic fields and measures far-field planar waves, enabling precise electrical structure imaging at depths of 50m-2km (Strangway et al., 1973). This technique has been extensively applied in mineral resource exploration, geological hazard evaluation, and engineering surveys (Guo et al., 2019; Fu et al., 2013; Hasan et al., 2025; Long et al., 2023).

However, the practical efficacy of CSAMT critically depends on instrument performance. Representative electromagnetic exploration instruments include the EH4/EH5 system (Geometrics, 2024) jointly developed by Geometrics and EMI (US), Metronix's ADU series (Germany) (Metronix, 2020), Zonge's GDP-32II (US) (Zonge, 2021), Phoenix's V8 system (Canada) (Phoenix, 2012), the SEP system by the Institute of Geology and Geophysics, CAS (Di et al., 2013, 2017), and Guoke Instrument's UltraEM Z4 magnetotelluric system (China) (Guoke, n.d.).

Although the performance of these instruments has reached a relatively high level and achieved engineering applications, there are still significant limitations in technical implementation: some instruments have a limited operating frequency band of 0.1Hz-10kHz and low sampling rates, resulting in insufficient resolution in shallow layers (

Figure 1. Structure diagram of the CSUMT-R system.

#### 2.1 Collection system structure

The collection system consists of three subsystems: the analog board, the connection board, and the interface board. Serving as the internal/external interaction hub of the instrument, the interface board integrates electrode/magnetic field sensor interfaces, 12V DC (Direct Current) power input, Gigabit Ethernet port, LED(Light Emitting Diode) status indicators, and operational buttons. Sensor signals are directly connected to the analog board's input channels through shielded cables to ensure low-noise transmission. Built-in overcurrent protection and battery monitoring circuits enable real-time power status monitoring through the main control board.

## 2.2 Software System Design

The software system mainly consists of two parts: the embedded control system and the upper-computer remote monitoring system. The upper-computer remote monitoring system runs on the Windows operating system platform and is programmed in C#. The embedded control system operates on the MPSoC (Multi-Processor System on Chip) platform of the main control

Figure 2. The CSUMT-R instrument.

board in the instrument and is programmed in C. The embedded control system can autonomously execute acquisition tasks or perform operations according to commands from the upper-computer monitoring system. The upper-computer monitoring system can remotely control the receiver via 4G (Fourth Generation) networks or establish a WLAN (Wireless Local Area Network) for on-site control.

#### 3 Embedded control system software design

This system is deployed on the Xilinx XCZU3EG heterogeneous MPSoC, serving as the core intermediary module between the upper-computer and the collection system. It handles critical functions including command forwarding, data storage, and instrument status monitoring. The system implements status monitoring, power management, LED control, and command transmission via the AXI4-Lite bus, utilizes SSD as the high-speed data storage medium, and relies on the embedded Linux operating system for task scheduling and resource management. Through foundational libraries (including time management, device drivers, disk operations, wireless networks, and upgrade interfaces), the system provides a unified HAL (Hardware Abstraction Layer) for upper-layer applications, supporting standardized cross-thread/process interface calls, with its underlying implementation dependent on Linux system calls and the GNU C Library. The overall software architecture is shown in Figure.

3. The system adopts a layered multi-thread architecture, achieving high concurrency through collaboration between the main thread and five functional sub-threads. The main thread executes core tasks such as data acquisition flow control and instrument calibration. The network communication thread implements WLAN/IoT communication based on TCP (Transmission Control Protocol)/UDP (User Datagram Protocol), supporting command interaction and data transmission with the upper-computer. The system status thread monitors critical parameters (SSD remaining capacity, battery voltage, instrument temperature, satel-

lite status, etc.) in real time and handles operational anomalies. The human-machine interaction thread indicates system status and warnings through LED states. The time management thread handles system time synchronization and dynamic correction. Core communication interface modules use message queues for inter-thread messaging, ensuring real-time and reliable data exchange.

Figure 3. Software structure diagram of device management system.

#### 95 3.1 Embedded control system workflow

100

105

After power-on, the system first executes initialization: loading drivers, configuring buffers, initializing wireless networks, allocating memory for global parameter structures, and initializing message queues. It then starts all threads. During initialization, the instrument performs self-testing. If self-test fails, the system generates logs based on error type and location, executes a safe shutdown, and enters standby mode for technician inspection. Post self-test, the instrument enters the main thread and automatically activates blind acquisition mode. In this mode, the system automatically configures acquisition parameters after GPS synchronization according to predefined frequency tables, generates JSON-format acquisition schedules, and sequentially executes acquisitions. After completing a cycle, the system automatically configures the next cycle based on the frequency table, ensuring continuous and convenient acquisition. Blind acquisition requires no upper-computer control or operator intervention, enabling immediate acquisition upon startup, significantly simplifying field operations and improving efficiency in complex environments. Concurrently, the system establishes a TCP server for listening and a UDP broadcast port to await scan packets. When the upper-computer discovers the receiver via UDP scan and establishes a TCP persistent connection, the receiver exits blind acquisition mode and enters manual control mode. In this mode, acquisition control is transferred to the upper-computer, with real-time data transmission. Connection status is monitored via heartbeat packets, and automatic reconnection is attempted upon disconnection.

### 110 4 Construction of Control System Operation Network

Harsh outdoor environments and large-scale remote station layouts may lead to data transmission issues. Therefore, building a suitable network is crucial to address these challenges and reduce the complexity of electromagnetic data acquisition in field operations.

For electromagnetic acquisition sites located in remote areas with unstable network conditions, practical deployment requires micro-WLAN arrays. We developed a distributed hybrid networking solution, as shown in Figure. 4. This solution combines wired and wireless connections: receivers connect to relay APs(Access Point) via wired or wireless links, relay APs connect wirelessly to the main AP, and monitoring terminals access the network through the central station's main AP. Using distributed networking, operators can control all receivers within a range using a single computer at the central station, improving efficiency.

Figure 4. WLAN networking topology.

The system uses 5.8 GHz WLAN for network access and data transmission. Receivers bind SSIDs(Service Set Identifier) via TCP, connect to relay APs via MAC(Media Access Control) addresses, and enforce MAC filtering for security. Each relay AP supports up to two receivers. Data is transmitted via 5.8 GHz WLAN to the central AP and then to monitoring terminals via network cables. The 5.8 GHz band was chosen for its high speed and anti-interference capabilities, ensuring stable and maintainable network topology.

For long-term electromagnetic signal acquisition in unmanned sites, we developed an IoT-based solution (see Figure. 5). This uses a 4G module (USR-G806w) for cloud communication, supporting LTE Cat4(UE-Category 4) (150 Mbps downlink, 50 Mbps uplink) in weak-signal areas. A star topology is built with a central router: receivers connect to sub-routers via Ethernet, and sub-routers use MQTT(Message Queuing Telemetry Transport) to link to the cloud. This low-power, plug-and-play design suits long-term multi-node acquisition (Qiao et al., 2020; Perdana et al., 2019; Ahmad et al., 2019; Zhou et al., 2021).

Figure 5. IoT networking topology.

145

In the process of network construction, first configure the central router with a static IP(Internet Protocol) and unique SSID, enabling DHCP(Dynamic Host Configuration Protocol). And the Sub-routers connect via WAN(Wide Area Network) ports, obtain the IP address, and register on the cloud; Receivers connect to sub-router via RJ45, receive private IPs, and bind device IDs(Identity)/GPS to the cloud; Remote upper-computers access via VPN(Virtual Private Network) using the central router's public IP address for cloud-based monitoring.

#### 135 5 Ultra-wide band data transmission technology

The system is based on CSUMT-R, with a dynamic range of more than 135 dB, a sampling rate ranging from 305 Hz to 2.5 MHz, and a data transmission speed ranging from 32 kbps to 320 Mbps. To address the data transmission challenges in ultra-wide band data acquisition, this study proposes a hierarchical high-speed data transmission and storage architecture, with the data transmission process illustrated in Figure. 6. The system workflow is divided into two coordinated phases: control command transmission and data acquisition transmission.

The control command transmission process is initiated by the upper-computer or the embedded control system. Acquisition parameters (sampling rate, gain, channel selection, etc.) are sent via the AXI4-Lite bus to the collection system main control module. The main control module employs a dual-buffer register design, supporting dynamic parameter updates and seamless state machine switching, ensuring efficient configuration of acquisition modes. The data transmission process is initiated by the collection system. After acquisition starts, the collection system constructs a five-channel parallel processing pipeline for the ADC. The data acquisition module receives data from the ADC and transmits it via the AXI4-Stream bus to the DMA module within the collection system. The DMA module processes the five-channel data and writes it into the DDR managed by the ARM. When the written data volume reaches a preset threshold, an interrupt signal is sent to the ARM. Upon receiving

Figure 6. Data transmission flowchart.

the interrupt signal, the ARM transfers the DDR data to the system-mounted SSD via the PCIe bus. If real-time monitoring is required, the data is simultaneously transmitted to the upper-computer via the TCP protocol, where it is analyzed, processed, and displayed as waveforms and spectra per channel.

A critical component of the data transmission process is the DMA driver. DMA technology, with its core feature of "bypassing the CPU," remains pivotal for enhancing system performance. Although traditional DMA transmission reduces CPU load, issues such as buffer management efficiency and interrupt response overhead become significant as data throughput demands surge.

The receiver generates data volumes ranging from 32 kbps to 320 Mbps under different sampling rates. Traditional DMA drivers use fixed-size buffers, which in low-rate scenarios may leave buffers unfilled for prolonged periods, causing transmission delays and memory waste, while in high-rate scenarios, insufficient buffer sizes may lead to data loss. To address this, we optimized the DMA driver by introducing a dynamic buffer adjustment mechanism, enabling an "elastic memory pool management" approach. During driver initialization, multiple pre-allocated buffers of varying sizes (256B–2MB) are created. At the start of acquisition, the driver selects an appropriate buffer based on the sampling rate. After acquisition begins, a built-in data rate monitoring module calculates the transmission rate using a sliding window. Combined with the STA/LTA algorithm, if the STA/LTA deviates from the threshold range, the buffer size is dynamically adjusted to ensure stable collection.

#### Two stages of dynamic buffer management

#### 1. Initial buffer allocation:

The baseline transmission rate is calculated as:

$$R_{\text{base}} = f_s \times \left(\frac{N_{\text{ch}} \times B_{\text{ADC}}}{8} + 1\right). \tag{1}$$

where  $f_s$  denotes the sampling rate (ranging from 305 Hz to 2.5 MHz),  $N_{\rm ch}$  represents the number of channels (5 channels),  $B_{\rm ADC}$  indicates the resolution (24-bit).

## 2. Runtime dynamic adjustment:

The dynamic buffer adjustment is triggered when the STA/LTA ratio exceeds the threshold range of [0.7, 1.3]. STA is calculated over a 1-second sliding window and LTA is calculated over a 10-second window. When the ratio is below 0.7, the buffer size is reduced by one level; when the ratio is greater than 1.3, it is increased by one level, until the STA/LTA stabilizes.

Additionally, the DMA driver allocates physically contiguous memory at the start of acquisition and directly maps the kernel buffer to user space. Users can access the DMA buffer directly without copying data via driver service functions, further reducing CPU overhead. The mapped virtual address region dynamically expands or contracts with buffer size changes, ensuring applications redirect to new memory regions during buffer switching. The dynamic buffer DMA driver is shown in Figure. 7.

Figure 7. Schematic diagram of dynamic buffer DMA driver.

After long-term on-site testing and verification(Section 7), the new DMA driver program did not experience data lag or packet loss during broadband data transfer and storage operations. This empirical verification proves the feasibility and robustness of our designed data transmission architecture under various on-site conditions.

#### 6 Upper-computer monitoring system software design

The upper-computer monitoring software adopts a C/S(Client/Server) architecture developed in C#, aiming to achieve network construction, real-time status monitoring of acquisition stations, batch control, real-time waveform display and spectrum analysis, local data viewing, and processing. The functional architecture of the upper-computer is shown in Figure. 8. The system primarily consists of five functional modules: data transmission and storage, real-time monitoring, real-time control, waveform display, and data preprocessing.

**Figure 8.** Functional architecture diagram of upper-computer monitoring software.

#### Functions of each module

200

205

- The data transmission and storage module handles real-time data transfer between the upper-computer and receivers, as well as local data uploads. It also supports software version updates and maintenance by logging into the receiver's file system via SSH (Secure Shell Protocol). The UDP protocol is responsible for network scanning and parsing status packets returned by receivers. After establishing a connection, it continuously sends heartbeat packets to monitor receiver status and evaluates communication integrity based on returned status information. The TCP protocol, chosen for its high reliability, congestion control, and full-duplex communication, is primarily used for real-time data transmission. Additionally, an SFTP (SSH File Transfer Protocol) client is integrated into the upper-computer software to access the receiver's file system, download acquisition data, and support software version maintenance and configuration updates, the transmission screenshot is shown in Figure. 9.
  - 2. The real-time monitoring module tracks real-time parameters of CSUMT-R, including device ID, operational status, synchronization status, battery voltage, battery current, and remaining storage space.
    - 3. The real-time control module manages receiver connection/disconnection and start/stop data acquisition. It also allows configuration of acquisition parameters such as electric and magnetic field channel gains and sampling rates.
    - 4. The waveform display module includes real-time and local data visualization. In real-time mode, users can adjust display point counts and FFT (Fast Fourier Transform) resolution to optimize computational precision. The system also supports work area, survey line/point configurations, displaying real-time waveforms, spectra, and apparent resistivity. For local data, the software reads and downloads receiver-stored data or local files on the computer, automatically detecting sampling rates and gains while applying calibration corrections to remove channel-specific effects. The waveform display interface design is shown in Figure. 10.

Figure 9. Self-developed upper-computer SFTP transmission screenshot.

**Figure 10.** Self-developed upper-computer waveform display interface design.

5. The data preprocessing module handles post-acquisition data processing for single or multiple survey points. It comprises a workspace configuration module and a data processing module, with the latter's interface shown in Figure. 11. Before processing, users create a workspace and define parameters such as workspace name, survey line/point names, electrode

spacing, channel selection, and calibration files. For CSAMT methods, the apparent resistivity formula is:

$$\rho_a = \frac{1}{\mu\omega} \frac{|E|^2}{|H|^2}.\tag{2}$$

and the phase formula is:

$$\phi = Arg\left(\frac{E}{H}\right). \tag{3}$$

Where  $\rho_a$  is apparent resistivity,  $\phi$  is phase,  $\mu$  is subsurface permeability,  $\omega$  is angular frequency, and E and H are measured electric and magnetic field intensities (Cagniard, 1953; Chen & Yan, 1995). The CSUMT-R captures two electric field channels  $(E_x, E_y)$  and three magnetic field channels  $(H_x, H_y, H_z)$ . The preprocessing module supports arbitrary channel selection, independent calibration files for all five channels, automatic gain calculation for electric/magnetic fields, and simultaneous computation of up to two apparent resistivity results. Results are saved in corresponding survey point lists. To enhance efficiency, the module enables batch processing of apparent resistivity and phase, randomly selecting 20 data segments for calculation and saving. Results with excessive errors are flagged for filtering. It also processes all data files from a single survey point, saving outputs uniformly. During processing, the module generates apparent resistivity-frequency and phase-frequency plots for visual trend analysis. Processed data can be exported as JSON or CSV files for further analysis, and JSON files can reload workspace configurations when reopening projects.

Figure 11. Self-developed upper-computer data preprocessing interface design.

After field testing and verification (Section 7), the developed software overcomes the obvious operational limitations of existing systems (compared to EH4 and V8):EH4 only displays apparent resistivity and phase diagrams without real-time waveform

visualization capabilities or raw channel data export functionality. Furthermore, neither V8 nor EH4 systems incorporate blind acquisition mode, which substantially increases operational time during repetitive multi-station surveys due to manual intervention requirements at each measurement point. In contrast, our system features a simplified and intuitive operation interface with automated acquisition capabilities, effectively reducing operator workload and improving field survey efficiency. These comparative tests validate that the developed software achieves substantial improvements in human-machine interaction complexity as designed.

#### 7 Instrument tests

To validate the consistency and reliability of the system's exploration results, we conducted comparative field tests in the Fengtai ore cluster area of Fengxian County, Shaanxi Province, China. The tests covered survey lines LS03 and LS05 (Figure. 12), each 3 km long with a 200 m spacing between lines, totaling 242 survey points at 25 m intervals. The transmission band was 1Hz–600kHz. The field station layout is shown in Figure. 13, with field photos in Figure. 14. The transmission module comprises a high-frequency transmitter(10kHz-600kHz) and a low-frequency transmitter(1Hz-10kHz). The low-frequency transmitter was positioned 8 km away from the vertical survey lines; preliminary tests verified that its transmitted signals could cover both survey lines, thus its position was maintained unchanged throughout the entire testing process. In contrast, according to the CSAMT method's skin depth formula:

$$\delta = \sqrt{\frac{2}{\mu_0 \sigma \omega}}.\tag{4}$$

Where  $\delta$  is skin depth,  $\mu_0$  is vacuum permeability,  $\sigma$  is medium conductivity, and  $\omega$  is angular frequency, when the Tx-Rx(Transmitter-Receiver) distance r is smaller than  $\delta$ , the electromagnetic field exhibits non-planar wave characteristics (near-field). To avoid near-field effects, the receivers were placed 200 m from the high-frequency transmitter. As confirmed by tests, the high-frequency signals can cover 7 measurement points (e.g., when the transmitter is situated at the LS05-105 measurement point, its transmitted signals can cover the range from LS03-99 to LS03-111). Therefore, the high-frequency transmitter was moved synchronously with the receiver to ensure that the receiver remained within the coverage of the transmitted signals.

To compare CSUMT-R with commercial instruments (V8 and EH4), considering the bandwidth of two instruments (V8:0.1Hz-10kHz,EH4:10Hz-92kHz), we conducted comparative tests: signals below 10 kHz were compared with V8, and 10 kHz-100 kHz signals with EH4. Results (Figure. 15, Figure. 16) show smooth data curves and high-quality signal acquisition, the apparent resistivity and phase curves are smooth in the high-frequency and low-frequency transmitter switching interval(10kHz). The mean square relative error between instruments was calculated as:

255 
$$m = \pm \sqrt{\frac{1}{2n} \sum_{i=1}^{n} \left(\frac{A_i - A_i'}{\overline{A}}\right)^2}.$$
 (5)

and the  $\overline{A}$  formula is

$$\overline{A} = \frac{A_i + A_i'}{2}.\tag{6}$$

Figure 12. Electromagnetic survey lines in Fengtai ore cluster.(©Google Earth)

Figure 13. Field station layout in Fengtai ore cluster.

Where n is the number of frequency points,  $A_i$  is apparent resistivity measured by CSUMT-R at the i-th frequency point,  $A_i'$  is apparent resistivity measured by the reference instrument or the second measurement by CSUMT-R,  $\overline{A}$  is mean of  $A_i$  and  $A_i'$  (Ministry of Land and Resources of the People's Republic of China, 2015). For CSUMT-R repeatability tests,  $A_i$  and  $A_i'$  represent two consecutive measurements at the same survey point. In all survey points, the mean square relative error of

Figure 14. Field test photos in Fengtai ore cluster.

CSUMT-R was below 3%, and the mean square relative error of CSUMT-R and commercial instruments was below 5%, which confirms the high reliability of the system. Figure. 17 displays the apparent resistivity pseudo-section for line LS05.

Figure 15. Self-developed upper-computer waveform display interface and data preprocessing interface during field validation in Fengtai.

## 8 Conclusions

This study developed a cloud and IoT-based ultra-high-frequency controllable-source electromagnetic receiver system software. The core innovations include: (1) a dynamic buffer DMA driver with elastic memory pools (256 B–2 MB) that adapts

Figure 16. Field data for apparent resistivity and impedance phase for the CSUMT-R and commercial instruments(V8 and EH4).

to sampling rates (305 Hz–2.5 MHz); (2) distributed hybrid networking integrating 5.8 GHz WLAN and 4G-IoT for multiscenario field deployment; and (3) five-channel batch processing algorithms supporting independent calibration and real-time visualization of ultra-wideband EM data. The main research content includes:

- Device management system design. A hierarchical multi-threaded architecture was implemented to enable efficient and stable remote data acquisition and transmission across diverse field scenarios. Field testing validates the system's capability to execute complex operational tasks seamlessly, maintain robust inter-system communication, and demonstrate advanced intelligence in electromagnetic exploration applications.
- 2. Upper-computer monitoring system design. The system integrates comprehensive functionalities including real-time data transmission, status monitoring, parameter configuration, waveform display, and data preprocessing for the CSUMT-R platform. Combined with apparent resistivity calculations and visual analysis based on the CSAMT method, the system provides a reliable intelligent solution for electromagnetic exploration in geologically complex environments.

Figure 17. Apparent resistivity pseudo-section for Line LS05 in Fengtai ore cluster.

Author contributions. This study was designed, tested, and implemented by the authors of the paper. The full text was designed and implemented by XZ. XZ and HL worked on the embedded control system design. XZ and ZL worked on the upper-computer monitoring system design. The other three authors(QZ, HZ, and XW) carried out revision and correction during the completion of the article. All authors contributed to the test.

Competing interests. The authors have no competing interests to disclose.

280

285

Acknowledgements. This study is supported by the Deep Earth probe and Mineral Resources Exploration-National Science and Technology Major Project (Grant No.2024ZD1000800, 2024ZD1000805), the National Key R&D Program of China (Grant No.2022YFF0706202 and No.2021YFC2801404), the National Natural Science Foundation of China (Grant NO.42074155) and the Key Research Program of the Chinese Academy of Sciences (Grant NO.KGFZD-145-22-06-02).

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
