# Peer review of "Software development of an internet-of-things based controlled-source ultra-audio frequency electromagnetic receiver"

_EGUsphere, 2025_

## Author Comment (AC1)

Dear reviewer,

We truly appreciate the time and energy you dedicated in carefully reviewing our manuscript. Your comments were highly helpful. We really appreciate your attention and comments on our manuscript. Our replies are listed as follows:

**Comments 1:**

Formula (4) for mean square relative error uses an undefined symbol $\bar{A}$;Include the formula for self-consistency checks of CSUMT-R apparent resistivity data.

**Response 1:**

Thank you for highlighting this ambiguity. We have revised Formula (4) and its description to explicitly define all symbols and clarify its dual use for both cross-instrument comparison and self-consistency validation:

Revised Formula (4):

$$m = \pm \sqrt{\frac{1}{2n} \sum_{i=1}^{n} \left( \frac{A_i - A_i^{'}}{\overline{A}} \right)^2} \tag{4}$$

and the $\overline{A}$ formula is

$$\overline{A} = \frac{A_i + A_i^{'}}{2} \tag{5}$$

(where $n$ is the number of frequency points, $A_i$ is apparent resistivity measured by CSUMT-R at the $i$-$th$ frequency point, $A_i^{'}$ is Apparent resistivity measured by the reference instrument or the second measurement by CSUMT-R, $\overline{A}$ is mean of $A_i$ and $A_i^{'}$).

Self comparison:

For CSUMT-R repeatability tests, $A_i$ and $A_i^{'}$ represent two consecutive measurements at the same survey point. The mean square relative error $m$ between repeated measurements was consistently <3%, confirming high instrument stability.

**Comments 2:**

The dynamic buffer DMA driver is a key innovation but lacks dedicated illustration. Replace Fig. 8 with a diagram explicitly detailing this mechanism.

**Response 2:**

We appreciate you raising this point. We have modified Fig. 8. During initialization, the DMA driver allocates memory space for the dynamic buffer. In the process of data transmission, the acquisition system writes data into the DDR. The DMA driver selects an appropriate buffer in real time according to the STA/LTA algorithm to ensure that data can be written into the SSD efficiently and accurately. The modified image is shown below:

[Figure]

Figure 8. Schematic diagram of dynamic buffer DMA driver.

**Comments 3:**

Add an opening paragraph summarizing the core innovations.

**Response 3:**

Thank you so much for your comment. The core innovations of this study are:

(1) A dynamic buffer DMA driver with elastic memory pools (256B–2MB) that adapts to sampling rates (305 Hz–2.5 MHz), eliminating data loss at 320 Mbps transmission;

(2) Distributed hybrid networking integrating 5.8 GHz WLAN and 4G-IoT for multi-scenario field deployment, enabling real-time remote monitoring in complex terrains;

(3) Five-channel batch processing algorithms supporting independent calibration and real-time visualization of ultra-wideband EM data.

These advancements collectively establish a robust framework for intelligent electromagnetic exploration in challenging environments.

**Comments 4:**

Unify the term for the monitoring software to "upper-computer" throughout (e.g., Section 6 uses "host").

**Response 4:**

Thank you for your careful review of the manuscript. All instances of "host" (including in Section 6) have been replaced with "upper-computer" for consistency.

---

## Author Response (AR1)

**Reply on CC1**

**Comments 1:**

Section 5 mentions buffer adjustment triggered when "STA/LTA deviates from the threshold range." To enable replication of the driver's optimization logic, explicitly define the STA/LTA thresholds (e.g., specific ratio bounds or values) and the sampling window sizes used for dynamic buffer adjustment.

**Response 1:**

We appreciate this suggestion for improving methodological transparency. *The following details have been added to Section 5 in the revised manuscript*:

The dynamic buffer management operates in two phases:

1. Initial buffer allocation:

The baseline transmission rate is calculated as:

$$R_{\text{base}} = f_s \times \left(\frac{N_{\text{ch}} \times B_{\text{ADC}}}{8} + 1\right)$$

where  $f_s$  = sampling rate (305 Hz–2.5 MHz),  $N_{\rm ch}$  = 5 channels,  $B_{\rm ADC}$  = 24-bit resolution. To ensure that the upper-computer has a refresh rate of 20 frames, the DMA buffer size is set as the baseline transmission rate divided by 20. When the sampling rate is 2.5 MHz, the buffer size is 2 MB, when the sampling rate is 305 Hz, the buffer size is 256B.

2. Runtime dynamic adjustment:

The dynamic buffer adjustment is triggered when the STA/LTA ratio exceeds the threshold range of [0.7, 1.3]. STA is calculated over a 1-second sliding window and LTA is calculated over a 10-second window. When the ratio is below 0.7, the buffer size is reduced by one level; when the ratio is greater than 1.3, it is increased by one level, until the STA/LTA stabilizes.

**Comments 2:**

Add a screenshot of the waveform display interface in the instrument testing section (Section 7).

**Response 2:**

Thank you so much for your comment. A screenshot of the real-time waveform display interface during field tests has been added in Section 7(Figure 15). The image is shown below:

Figure 15. Self-developed upper-computer waveform display interface during field validation in Fengtai.

**Comments 3:**

Section 6 introduces the design of an SFTP-based file transfer function in the upper-computer software. To substantiate the implementation and user interaction, provide a screenshot of the software's SFTP transfer interface in action or a test result screenshot confirming successful transfer.

**Response 3:**

Your suggestion is highly valued. We have included a new Figure 9 in Section 6 demonstrating the SFTP functionality. The SFTP interface consists of an upper-computer section, a CSUMT-R section, and a progress bar. Both the upper-computer and CSUMT-R can select paths, and the upload and download between them are bidirectional. It allows not only the retrieval of collected data but also the system maintenance and upgrading of CSUMT-R. If the upper-computer is connected to multiple CSUMT-R devices, the desired one can be selected through a drop-down box, and the progress bar will display the upload or download progress in real time.

Figure 9. Self-developed upper-computer SFTP transmission screenshot.

**Comments 4:**

Figure. 7 attempts to illustrate both command transmission and data transmission flows. Currently, these distinct processes are visually intertwined, making the diagram difficult to interpret. Please clearly differentiate the command flow from the data flow within this figure.

**Response 4:**

We appreciate you raising this point. Figure 7 has been redesigned to decouple the two workflows: command flow (blue arrows) and data flow (red arrows). The command flow traverses TCP/UDP for network control to the ARM core, then AXI4-Lite bus to configure registers; concurrently, the data flow pipes raw ADC samples via AXI4-Stream to DMA, then through PCIe to SSD storage or via TCP for real-time display. The modified image is shown below:

Figure 7. Data transmission flowchart.

**Comments 5:**

Revise the phrase "cloud- and IoT-based" to "cloud and IoT-based" (Section 8) to maintain grammatical precision in describing the software architecture.

**Response 5:**

Thank you for your careful review of the manuscript. We have modified the sentence.

**Reply on RC1**

**Comments 1:**

Formula (4) for mean square relative error uses an undefined symbol  $\bar{A}$ ; Include the formula for self-consistency checks of CSUMT-R apparent resistivity data.

**Response 1:**

Thank you for highlighting this ambiguity. We have revised Formula (4) and its description to explicitly define all symbols and clarify its dual use for both cross-instrument comparison and self-consistency validation:

Revised Formula (4):

$$m = \pm \sqrt{\frac{1}{2n} \sum_{i=1}^{n} \left(\frac{A_i - A_i'}{\overline{A}}\right)^2}$$
 (5)

and the  $\overline{A}$  formula is

$$\overline{A} = \frac{A_i + A_i'}{2} \tag{6}$$

(where n is the number of frequency points,  $A_i$  is apparent resistivity measured by CSUMT-R at the *i-th* frequency point,  $A'_i$  is Apparent resistivity measured by the reference instrument or the second measurement by CSUMT-R,  $\overline{A}$  is mean of  $A_i$  and  $A'_i$ ).

Self comparison:

For CSUMT-R repeatability tests,  $A_i$  and  $A'_i$  represent two consecutive measurements at the same survey point. The mean square relative error m between repeated measurements was consistently

Figure 8. Schematic diagram of dynamic buffer DMA driver.

**Comments 3:**

Add an opening paragraph summarizing the core innovations.

**Response 3:**

Thank you so much for your comment. The core innovations of this study are:

- (1) A dynamic buffer DMA driver with elastic memory pools (256B–2MB) that adapts to sampling rates (305 Hz–2.5 MHz), eliminating data loss at 320 Mbps transmission;
- (2) Distributed hybrid networking integrating 5.8 GHz WLAN and 4G-IoT for multi-scenario field deployment, enabling real-time remote monitoring in complex terrains;
- (3) Five-channel batch processing algorithms supporting independent calibration and real-time visualization of ultra-wideband EM data.

These advancements collectively establish a robust framework for intelligent electromagnetic exploration in challenging environments.

We added a summary of innovative points and optimized the research content in Section 8.

**Comments 4:**

Unify the term for the monitoring software to "upper-computer" throughout (e.g., Section 6 uses "host").

**Response 4:**

Thank you for your careful review of the manuscript. All instances of "host" (including in Section 6) have been replaced with "upper-computer" for consistency.

**Reply on RC2**

**1) Emphasis on Problem-Solving in Abstract/Introduction**

**Comments 1:**

The abstract highlighted targeted improvements in: (i) shallow-to-medium layer exploration accuracy, (ii) human–machine interaction complexity, and (iii) constrained data transmission. Suggest strengthening comparative analysis in the main text.

**Response 1:**

We sincerely appreciate the reviewer's constructive suggestion regarding the comparative analysis.

In response, we have strengthened the comparison between our system and commercial instruments in the main text by incorporating the following clarifications:

i) Shallow-to-medium layer exploration accuracy (Section 1):

We added explanatory text in Section 1 highlighting that our instrument can acquire signals at frequencies up to 1 MHz, whereas commercial instruments (e.g., EH4) are limited to 100 kHz. According to the CSAMT (Controlled Source Audio-frequency Magnetotelluric) methodology, this extended frequency acquisition capability enables our system to achieve enhanced resolution for shallow subsurface structures, thereby addressing the targeted improvement in shallow-to-medium layer exploration mentioned in the abstract.

ii) Human-machine interaction complexity (Section 6):

At the end of Section 6, we supplemented the text with comparative observations from field testing. Specifically, we noted that the EH4 commercial system only displays apparent resistivity and phase diagrams without real-time waveform visualization capabilities or raw channel data export functionality. Furthermore, neither the V8 nor EH4 systems incorporate blind acquisition mode, resulting in substantially increased operational time during repetitive multi-station surveys. These limitations contrast with our system's simplified and intuitive operation interface and automated acquisition features, validating the interaction complexity improvements cited in the abstract.

iii) Constrained data transmission (Section 5):

At the end of Section 5, we added descriptive text confirming that throughout extensive field testing, our instrument exhibited no data stuttering or packet loss during high-speed data transmission and storage operations. This empirical validation demonstrates the feasibility and robustness of our designed data transmission architecture under various field conditions.

**2) Clarify Field Test Configuration**

**Comments 2:**

Provide detailed descriptions of the 242 measurement points (distribution, geophysical parameters). Enhance Figure 15 by including physical images of EH4 instruments for direct comparison.

**Response 2:**

Thank you so much for your comment. We have modified and expanded the text of survey point description in Section 7. Brief information regarding the 242 measurement points was presented in the original manuscript. Herein, supplementary details of the measurement points and testing procedure are provided as follows:

Each survey line has a length of 3 km, with an interval of 200 m between the two parallel survey lines. A total of 242 measurement points were deployed, with an adjacent point spacing of 25 m; each survey line contains 121 measurement points. The layout of field test stations is illustrated in Figure 14.

The transmission module comprises a high-frequency transmitter and a low-frequency transmitter. The low-frequency transmitter was positioned 8 km away from the vertical survey lines; preliminary tests verified that its transmitted signals could cover both survey lines, thus its position was maintained unchanged throughout the entire testing process. In contrast, the high-frequency transmitter was placed 200 m away from the survey lines. As confirmed by tests, the high-frequency signals can cover 7 measurement points (e.g., when the transmitter is situated at

the LS05-105 measurement point, its transmitted signals can cover the range from LS03-99 to LS03-111). Therefore, the high-frequency transmitter was moved synchronously with the receiver to ensure that the receiver remained within the coverage of the transmitted signals. *During the testing process, the station layout at each measurement point was consistent with the receiver layout diagram depicted in Figure 13*.

*Figure 14 has been updated;* the revised field test photographs include the CSUMT-R, EH4, and V8 instruments for visual comparison.

Figure 13. Field station layout in Fengtai ore cluster.

Figure 14. Field test photos in Fengtai ore cluster.

**3) Validation of Formula 4 (Mean Square Relative Error) Comments 3:**

Clarify whether the metric is industry-standard; cite authoritative sources or explain rationale if novel. Provide contextual validation.

**Response 3:**

We appreciate you raising this point. We have added source information after this formula in Section 7. This formula, designed to assess the consistency of two or more receivers, is derived from Technical Specification for Controlled-Source Audio-Frequency Magnetotellurics (Standard No.: DZ/T 0280-2015)—a national technical standard widely recognized in the Chinese geophysical exploration industry. As stipulated in this standard, prior to initiating fieldwork in the same survey area, a consistency verification test must be conducted for two or more receivers of the same model. The key requirements of this test are specified as follows:

- a) The consistency verification of instruments shall be performed under actual field conditions. A site with minimal electromagnetic interference shall be selected to conduct single-point, full-frequency-band measurements.
- b) The consistency of the instruments shall be quantified using the total mean square relative error  $\varepsilon$  of the Cagniard resistivity, as observed by m instruments at a given measurement point. The calculation formula specified in the standard is:

$$\varepsilon = \pm \sqrt{\sum_{i=1}^{n} \sum_{j=1}^{m} V_{ij}^{2} / (L - n)}$$

Where  $V_{ij}$  denotes the relative error between the Cagniard resistivity value observed by the j-th instrument at the i-th frequency point and the average Cagniard resistivity value of m instruments at the i-th frequency point; mrepresents the number of instruments involved in the consistency verification test; L is the total count of relative errors  $V_{ij}$ , with  $L = m \times n$ ; n stands for the number of observation frequency points included in the consistency verification test.

The formula employed in this paper is a simplified version of the aforementioned standard formula, with the simplification process ensuring no loss of calculation accuracy for the specific application scenario of this study.

**4) Robustness of Instrument Comparison**

**Comment 4:**

Figure 17 shows results from a single commercial instrument. Add 1–2 additional instruments (e.g., V8, EH4) to enhance statistical credibility.

**Response 4:**

We have revised Figure 17, now Figure 17 is numbered as Figure 16. The updated figure now includes apparent resistivity plots and phase plots showing the comparisons between CSUMT-R and EH4, as well as between CSUMT-R and V8. Additionally, it incorporates a full-frequency-band comparison plot.

Figure 16. Field data for apparent resistivity and impedance phase for the CSUMT-R and commercial instruments.

**Future Recommendations**

**1) Enhanced Field Resilience (BeiDou Satellite Messaging)**

**Comments 1:**

Integrate BeiDou satellite messaging for data transmission in remote/extreme environments.

**Response 1:**

We thank the reviewer for this insightful suggestion. We have recognized the significant potential of BeiDou satellite messaging for data transmission under challenging field conditions, particularly in remote areas with limited conventional communication infrastructure. The integration of BeiDou satellite communication capabilities into our system is currently under active investigation.

**2) Long-Term Reliability Testing**

**Comments 2:**

Include 6–12 month unattended operation data to assess sustained reliability.

**Response 2:**

We sincerely appreciate the reviewer's emphasis on long-term reliability assessment, which is crucial for field geophysical instrumentation. In response to this suggestion, we have conducted comprehensive reliability testing of our system. From April 2025 to October 2025, field tests were performed in Shaanxi Province, China. During this 6-month deployment period, the system

successfully completed a rigorous 3,000-hour fault-free operation test under various environmental conditions, including temperature fluctuations, humidity variations, and electromagnetic interference. This extended reliability testing has been independently verified and certified by a third-party testing institution, confirming the system's sustained operational stability and robustness.

**3) Methodological Expansion (MT, IP)**

**Comments 3:**

Extend applicability to MT (deep imaging) and IP (mineral exploration).

**Response 3:**

We thank the reviewer for highlighting the importance of methodological versatility. We would like to clarify that our current system already supports magnetotelluric (MT) functionality, including real-time waveform visualization, time-series data processing, and the generation of apparent resistivity and phase pseudosections along survey profiles. These capabilities enable effective deep subsurface imaging applications. Regarding induced polarization (IP) methods for mineral exploration applications, we fully acknowledge their significance and plan to extend our research to encompass IP and other geophysical techniques in future work. This expansion will require additional hardware modifications and algorithm development, which are scheduled as part of our ongoing research program.

**Other changes**

- 1. We have modified the dimensions of some of the images to better fit them into the text;
- 2. We have *removed Figure 3 from the original manuscript* because we believe it is not beneficial for explaining the system;
- 3. We have added a funding information based on the following reasons. The initial work of this study (including theoretical analysis and preliminary experiments) was mainly supported by the National Key R&D Program of China (Grant No. 2022YFF0706202 and No. 2021YFC2801404). During the later stage of the study, the Deep Earth Probe and Mineral Resources Exploration National Science and Technology Major Project (Grant No. 2024ZD1000800, 2024ZD1000805) provided important support for the critical long-term testing and data verification, which was completed after the first draft of the manuscript. We would like to express our sincere gratitude for this support.